# Biologic Impact of Green Synthetized Magnetic Iron Oxide Nanoparticles on Two Different Lung Tumorigenic Monolayers and a 3D Normal Bronchial Model—EpiAirway^TM^ Microtissue

**DOI:** 10.3390/pharmaceutics15010002

**Published:** 2022-12-20

**Authors:** Elena-Alina Moacă, Claudia Watz, Alexandra-Corina Faur, Daniela Lazăr, Vlad Socoliuc, Cornelia Păcurariu, Robert Ianoș, Cristiana-Iulia Rus, Daliana Minda, Lucian Barbu-Tudoran, Cristina Adriana Dehelean

**Affiliations:** 1Faculty of Pharmacy, “Victor Babes” University of Medicine and Pharmacy Timisoara, 2nd Eftimie Murgu Square, RO-300041 Timisoara, Romania; 2Research Centre for Pharmaco-Toxicological Evaluation, “Victor Babes” University of Medicine and Pharmacy, 2nd Eftimie Murgu Square, RO-300041 Timisoara, Romania; 3Faculty of Medicine, “Victor Babes” University of Medicine and Pharmacy Timisoara, 2nd Eftimie Murgu Square, RO-300041 Timisoara, Romania; 4Center for Fundamental and Advanced Technical Research, Laboratory of Magnetic Fluids, Romanian Academy—Timisoara Branch, 24 M. Viteazu Ave., RO-300223 Timisoara, Romania; 5Research Center for Complex Fluids Systems Engineering, Politehnica University of Timisoara, 1 M. Viteazu Ave., RO-300222 Timisoara, Romania; 6Faculty of Industrial Chemistry and Environmental Engineering, Politehnica University Timisoara, 2 Victoriei Square, RO-300006 Timisoara, Romania; 7Electron Microscopy Laboratory “Prof. C. Craciun”, Faculty of Biology & Geology, “Babes-Bolyai” University, 5-7 Clinicilor Street, RO-400006 Cluj-Napoca, Romania; 8Electron Microscopy Integrated Laboratory, National Institute for R & D of Isotopic and Molecular Technologies, 67-103 Donat Street, RO-400293 Cluj-Napoca, Romania

**Keywords:** green co-precipitation synthesis, *Camellia sinensis*, *Ocimum basilicum*, magnetite, maghemite, A549, NCI-H460, EpiAirway^TM^

## Abstract

The present study reports the successful synthesis of biocompatible magnetic iron oxide nanoparticles (MNPs) by an ecofriendly single step method, using two ethanolic extracts based on leaves of *Camellia sinensis* L. and *Ocimum basilicum* L. The effect of both green raw materials as reducing and capping agents was taken into account for the development of MNPs, as well as the reaction synthesis temperature (25 °C and 80 °C). The biological effect of the MNPs obtained from *Camellia sinensis* L. ethanolic extract (Cs 25, Cs 80) was compared with that of the MNPs obtained from *Ocimum basilicum* L. ethanolic extract (Ob 25, Ob 80), by using two morphologically different lung cancer cell lines (A549 and NCI-H460); the results showed that the higher cell viability impairment was manifested by A549 cells after exposure to MNPs obtained from *Ocimum basilicum* L. ethanolic extract (Ob 25, Ob 80). Regarding the biosafety profile of the MNPs, it was shown that the EpiAirway^TM^ models did not elicit important viability decrease or significant histopathological changes after treatment with none of the MNPs (Cs 25, Cs 80 and Ob 25, Ob 80), at concentrations up to 500 µg/mL.

## 1. Introduction

According to Globocan’s estimates [1], lung cancer represents the second most commonly diagnosed tumor (one in 10 cancers diagnosed), with 2.2 million new cases in 2020, and the leading cause of cancer death worldwide (one in 5 cancer deaths), with approximately 1.8 million deaths. In Romania in 2020, lung cancer ranks the first place for incidence and mortality rates among all cancer sites, totalizing 12,122 new cases (12.3%) and 10,779 deaths (19.8%), equating 4.33% of total deaths.

Histologically, lung cancer can be classified into Small Cell Lung Cancer (SCLC) or Non-Small Cell Lung Cancer (NSCLC) (subdivided into several subtypes including lung adenocarcinoma), with the latest being the most common pathological type (about 85% of cases) [2,3,4]. Classical treatment strategies of lung cancer include surgery, chemotherapy/radiation therapy; in addition, metastatic disease treatment options include also targeted therapy and immunotherapy [5]. Because chemotherapy is less able to enrich the tumor area, several strategies to overcome these drawbacks were investigated, such us: prodrug administration with enhanced drug release in the cytoplasm of the tumor cell and nanotechnology-enabled formulations. In addition, employing nanoparticle platforms loaded with chemotherapeutic agents leads to a prolonged circulation time of the drugs and a facilitated target of the tumor site due to the enhanced permeability and a retention (EPR) effect [6]. Moreover, several studies investigated the capacity of nanomedicine to induce not only cancer-specific cytotoxicity, but also to target tumor microenvironment to optimize the nano-based therapy [7]. However, the tumor heterogeneity makes classic chemotherapies difficult to function; thus, a new pathway is emerging by using a different therapeutic approach through personalized medicine approached, thus taking advantage of both pharmacogenetics and nanomedicine [8].

Nowadays, the nanosystems based on functionalized magnetic nanoparticles (MNPs) offer a great promise for targeted lung cancer therapy, especially for NSCLC [9]. The smart MNPs are designed to target key molecules in order to curb tumor growth, metastasis and/or cell proliferation, resulting in enhanced pharmacological effects such as biocompatibility, low toxicity, and bioavailability in the human body [9,10]. The MNPs, are used for their unique and tailorable features, especially for their strong magnetic moment and superparamagnetic behavior, two paramount requirements for targeted NSCLC therapy as well as for magnetic hyperthermia—an alternative cancer treatment approach [11]. In addition, among the desired features of magnetic nanoparticles suitable for cancer therapy, there are also the pure chemical composition, the nanoscale size (under 100 nm)—which allow enhanced circulation and precise distribution, chemical stability in various aqueous and/or biological environments, as well as the surface chemistry of the nanoparticles, strongly dependent on the pH value, which can be modified and functionalized by grafting with various surfactants/biological ligands. 

In the last decades, significant research results have been reported regarding the application of Fe_3_O_4_ and/or γ-Fe_2_O_3_ in NSCLC targeted therapy [12,13,14,15,16,17,18]. To date, it is certain that these studies show both promising results regarding the use of MNPs for lung cancer diagnostic [19] and also their valuable features as nanocarriers for anticancer drug delivery to the lungs [20]; however, they are not yet accepted as an efficient clinical approach for humans. The studies reported above describe the use of MNPs obtained by chemical or physical methods. Nevertheless, the scientific literature presents few reports regarding the fabrication of magnetic iron oxides nanoparticles through green synthesis approach with anticancer potential against lung cancer [21,22,23]. No study has previously reported the use of basil as plant material for MNPs’ biosynthesis, whereas only one study reports the MNPs’ biosynthesis using green tea extract, as well as the cytotoxic effect of this MNPs against A549-luc-C8 non-small lung cancer line [24]. However, in that study [22], the authors did not investigate the cytotoxic effect of naked MNPs (obtained as such from green synthesis), they loaded the MNPs obtained with doxorubicin and curcumin and afterwards, the cytotoxic effect of the nanoformulation against NSCLC line was investigated. Nevertheless, it is well known that doxorubicin is a chemotherapeutic agent often used in cancer therapy, especially in lung tumors and curcumin is a bioactive compound with anticancer properties. 

Therefore, the aim of the present study consists in the biosynthesis of MNPs using *Camellia sinensis* L. (green tea) and *Ocimum basilicum* L. (basil) as plant materials, as well as the investigation of anticancer potential of the *as such* MNPs (uncoated with chemotherapeutic agent), against lung cancer—a topic never investigated before, which consists in the novelty of the present study. In order to determine the anticancer potential of the MNPs, the biological impact of the synthesized MNPs through the green method was evaluated. The biosafety profile of the newly synthesized MNPs was assessed using the EpiAirway^TM^ model (MatTek Corporation)—three-dimensional (3D) functional microtissues obtained from normal bronchial cells, cultured at the air–liquid interface. The 3D microtissues present a ciliated apical surface, active mucin secretion and high similarity with in vivo respiratory tract [25,26,27]. In addition, the anti-tumorigenic activity of the MNPs was assessed on two morphologically different lung cancer 2D cell lines: (i) human lung carcinoma—A549 cells and (ii) large human lung carcinoma—NCI-H460 cells, by using different in vitro techniques (Almar blue test and LDH method). 

## 2. Materials and Methods

### 2.1. Materials

For the synthesis of both ethanolic extracts, ethanol (≥99.8%, p.a.) acquired from Carl Roth Company (Karlsruhe, Germany) was used. For the biosynthesis of magnetic nanoparticles, a mixture of aqueous iron salts (FeCl_3_·6H_2_O FeSO_4_·7H_2_O), both purchased from Merck (Darmstadt, Germany) were used. In order to obtain nanoparticles with a pure chemical composition (i.e., Fe_3_O_4_), the Fe^3+^:Fe^2+^ molar ratio used was 2:1. The precipitation was carried out using NH_4_OH 25%, acquired from Chemical Company SA, Iasi, Romania. Ultrapure water from Milli-Q^®^ Integral Water Purification System (Merck Millipore, Darmstadt, Germany) was used to prepare the 70% ethanol solution and to dissolve the iron salts used as precursors. All the reagents were of analytical grade and used without any further purification.

### 2.2. Plant Collection and Extracts Preparation Protocol

*Camellia sinensis* L. (batch no. 10/21) and *Ocimum basilicum* L. (batch no. 11/21) plant materials were purchased from a local authentic herbal distributor—Favisan SRL (Lugoj, Romania). The green tea ethanolic extract was prepared according to the slightly modified protocol of Perva-Uzunalic et al. [28]. Briefly, 10 g of green tea leaves was grounded until a fine powder was obtained. The extraction process was done using the Soxhlet extraction, at 70 °C. The extraction was repeated three times (1 h/cycle), using a total volume of 200 mL ethanol 99.8%. After the three successive extractions, the extract was passed through a 0.45 µm Whatman membrane filters (Sigma–Aldrich, St. Louis, MO, USA; Merck KGaA, Darmstadt, Germany) and concentrated under reduced pressure at 35 °C, using a Heidolph G3 rotary evaporator (Schwabach, Germany) and then lyophilized in a Christ Alpha 1–2 freeze dryer (Osterode, Germany) at −60 °C. The basil extract was prepared according to the slightly modified protocol described by our research group in a previous work [29]. Briefly, 10 g of basil leaves hand-grounded, were mixed with 100 mL 70% ethanol and after a 15 min rest, the sample was subjected to ultrasound bath for 30 min, at 50 °C and 40 KHz (LBS2 10 L from FALC Instruments, Bergamo, Italy). After that, the extract was filtered through a 0.45 µm Whatman membrane filter, concentrated under reduced pressure at 50 °C, and lyophilized at −60 °C. The obtained powders were stored at 4 °C in a glass container until further use.

### 2.3. Fabrication Procedure of MNPs by Green Synthesis

According to the protocol of Abdullah and co-workers [30], the lyophilized powders of plant extracts were dissolved in 50 mL ethanol 99.8%, until a final concentration of 6 mg/mL was obtained. The ethanolic extracts were subjected separately to magnetic stirring at 450 rpm (Heidolph™ Hei-Tec Magnetic Stirrer Hotplates (Heidolph Instruments GmbH & Co. KG, Schwabach, Germany)), at different temperatures (25 °C and 80 °C). When the set temperatures were reached in the reaction medium, the metallic precursor aqueous solution was added (50 mL). After 1 h, 25 mL of NH_4_OH 25% was added dropwise to the metal precursor solution and plant extract, and the whole mixture was left under magnetic stirring for another 1 h. After all, the samples were cooled down to ambient temperature, the black precipitates obtained, strongly attracted by the neodymium block magnet (Model: Q-60-30-15-N, Webcraft GmbH, Gottmadingen, Germany) were washed three times with ultrapure water and dried at 70 °C in a POL-EKO oven (Wodzisław Slaski, Poland), under air atmosphere, for 24 h. After drying, all the samples were transferred to porcelain dishes and hand-ground until a fine black magnetic powder was obtained (Table 1).

### 2.4. Characterization Techniques of the Formed MNPs

The preformed magnetic iron oxide nanoparticles were subjected to physicochemical characterization for the establishment of the phase composition (through X-ray diffraction (XRD)), the MNPs stability with temperature (thermal behavior (TG-DSC)), particle size, morphology, and elemental composition of MNPs (through electronic microscopy—transmission (TEM) and scanning (SEM)), as well as the magnetic behavior in a magnetic field.

#### 2.4.1. Powder X-ray Diffraction (XRD)

In order to investigate the phase composition of the synthesized MNPs, each sample was subjected to X-ray diffraction (XRD), using the Rigaku Ultima IV diffractometer (Tokyo, Japan), with monocromated CuKα radiation. A voltage of 40 kV and a current of 40 mA were the set parameters. The Debye–Scherer Equation (1) was used to calculate the crystallite size of the MNPs, according to the most intense peak (311 hkl plane).
(1)$$D_{XRD}=\frac{0.9\cdot \lambda }{\beta \cdot \cos\theta }$$
where *D_XRD_* is the crystallite size of the MNPs (nm), *λ* is the radiation wavelength (0.15406 nm), *β* is the full width at half of the maximum (radians), and *θ* is the Bragg angle (degree).

#### 2.4.2. Thermal Behavior of the Synthesized MNPs

The thermal behavior of the synthesized MNPs was studied in the temperature range of 25–700 °C, using the Netzsch STA 449C instrument (Selb, Germany), equipped with alumina crucibles. The parameters set for the thermogravimetric (TG) and the differential scanning calorimetry (DSC) curves were: 10 °C/min heating rate under a flow rate of 20 mL/min and an artificial air atmosphere.

#### 2.4.3. Electron Microscopy Investigations

In order to investigate the surface morphology and the elemental composition (qualitative and semi-quantitative analysis) of the synthesized MNPs, the cold field emission- SEM was performed. The analysis was carried out using the Hitachi SU8230 cold field emission gun STEM (Chiyoda, Tokyo, Japan) microscope, equipped with EDX detectors X-Max^N^ 80 from Oxford Instruments (Bristol, UK). The MNPs were sputter-coated with carbon, mounted on a copper grid support and the analysis proceeded at an acceleration voltage of 30 kV in a high vacuum mode. The elemental composition of the MNPs was assessed by EDX analysis, which identifies the elements present in sample and expresses them in weight percent (wt%). 

The MNPs size and shape were assessed by TEM. The analysis was performed using the Hitachi HD2700 cold field emission gun STEM microscope (Chiyoda, Tokyo, Japan), with two windowless EDX detectors (X-Max^N^ 100). The samples were placed on a carbon-coated copper grid support and analyzed after drying. At 200 kV acceleration voltage, the images were obtained. 

#### 2.4.4. Magnetic Properties of MNPs

The magnetization of the MNPs was measured by vibrating sample magnetometry (VSM) at room temperature in 0–1000 kA/m magnetic field range, using a VSM 880 mag-netometer (ADE Technologies, Westwood, MA, USA). The initial magnetization curves were used to determine the MNPs’ magnetic diameter (*D_m_*) statistics by means of magne-togranulometry [31]. The MNPs’ saturation magnetization (M_sat_), remanent magnetization (M_r_), and coercive field (H_c_) were obtained from the hysteresis curves.

As the methodology section include a multidue of methods, in order to facilitate the overall understanding of the manuscript, a graphical abstract of the methodology section is presented below in Figure 1.

### 2.5. In Vitro Evaluations

#### 2.5.1. Culture Procedure

The present study was conducted using two different lung carcinoma cell lines, as follows: (i) A549 cells (CCL-185 from ATCC, LGC Standards GmbH)—lung carcinoma with an epithelial-like morphology; (ii) NCI-H460 (HTB-177 from ATCC, LGC Standards GmbH) cell line—large lung cancer cells. Both cell lines were stored under standard conditions (vapor phase of liquid nitrogen) before culturing.

The EpiAirway 3D in vitro microtissues (Air-100, Lot 33251, Kit C) were acquired from MatTek Life Science Company (Bratislava, Slovak Republic) and were handled according to the EpiAirway^TM^ AIR-100 Use Protocol. 

More details regarding the culture conditions are presented in the Appendix A.

#### 2.5.2. Biosafety Profile Using the EpiAirway^TM^ 3D In Vitro Microtissues through MTT Test

The tissue inserts were treated with 50 µL of test samples for 24 h. A detailed protocol is presented in the Appendix A.

The viable tissues converted the MTT reagent to a purple formazan salt; thus, the intensity of the purple color is proportional to the viability of the tissue. To quantify the viability percentage, the optical density (O.D.) of each well was spectrophotometrically determined at two different wavelengths (570 nm and 650 nm) and the following formula was used (Equation (2)):(2)$$EpiAirway\;Viability\;(\%)=\frac{{O.D.}_{{sample}_{570}}-{O.D.}_{{sample}_{650}}}{{O.D.}_{{control}_{570}}-{O.D.}_{{control}_{650}}}\times 100$$

#### 2.5.3. Histopathological Assessment of 3D EpiAirway^TM^ In Vitro Microtissues

For histopathological analysis, the 3D respiratory tissue models were fixed in 10% formalin, paraffin-embedded and then sectioned. The 4 µm thick formalin-fixed paraffin-embedded tissue samples were stained with hematoxylin and eosin (HE). The obtained slides were studied and photographed using the Leica DM750 microscope with digital camera (with magnification of 40× and objectives (×40 Ob.)). 

#### 2.5.4. Cell Viability Assessment by Means of Alamar Blue Colorimetric Test

The Alamar blue test is a well-known colorimetric method that quantifies the viable cell population based on their ability to transform resazurin (a blue compound), through functional mitochondrial reductase to resorfin (a pink compound) [32].

To quantify the effect induced by test samples, the Alamar blue colorimetric assay was performed using a previously published protocol [33], by measuring the absorbance of each well after addition of the Alamar blue reagent at two different wavelengths (570 nm and 600 nm) using a microplate reader (xMark^TM^ Microplate, Bio-Rad Laboratories, Hercules, CA, USA) and employing a previously published formula [34]. More details regarding this protocol are presented in the Appendix A.

#### 2.5.5. Cytotoxicity Assay by Quantifying the Lactate Dehydrogenase (LDH) Released

To quantify the LDH release into the extracellular medium, the absorbance of each well was measured spectrophotometrically at two different wavelengths (490 nm and 680 nm) by using a microplate reader (xMark^TM^ Microplate, Bio-Rad Laboratories, Hercules, CA, USA). The entire protocol is presented in the Appendix A.

### 2.6. Statistical Analysis

GraphPad Prism 5 version (GraphPad Software, San Diego, CA, USA) was employed for data representation and statistical analysis. Results are presented as mean of three independent experiments (n = 3) ± standard deviation (SD). One-way ANOVA was used to obtain the statistical differences, followed by Dunnett’s post-test.

## 3. Results 

### 3.1. Fabrication and Physicochemical Characterization of the Test Compounds by Green Synthesis

Starting from two ethanolic extracts based on leaves of *Camellia sinensis* (Cs) and *Ocimum basilicum* (Ob) and employing the green synthesis method, two types of MNPs were obtained. In order to investigate the influence of temperature on the features of the synthesized MNPs (size, shape, phase composition, biocompatibility, and stability), two different temperatures (25 °C and 80 °C) were employed. Figure 2 shows the main phytocompounds in green tea and basil leaves ethanolic extracts, which act both as reducing and capping agents during synthesis of MNPs.

#### 3.1.1. MNPs’ Phase Composition

Figure 3 presents the XRD patterns of the MNPs obtained from leaves ethanolic extracts of *Camellia sinensis* (Cs) and *Ocimum basilicum* (Ob), at 25 °C and 80 °C, using NH_4_OH 25% as precipitation agent.

The marked diffraction peaks of the obtained MNPs are very similar and correspond to the inverse spinel structure of cubic magnetite, according to the characteristic of Fe_3_O_4_ NPs pattern (PDF: 190629). The diffraction peaks of the resulted MNPs are located at 2*θ* values around 18.18°; 30.1°; 35.49°; 43.16°; 53.47°; 57.12°; 62.72°; 71.07°; 74.22°, which correspond to the following Bragg’s planar reflections: (111), (220), (311), (400), (422), (511), (440), (533), and (444). According to Table 2, the 2*θ* values of the resulted diffraction peaks from the XRD analysis are more appropriate to the 2*θ* values of the magnetite diffraction peaks than maghemite (PDF: 391346). Based on this statement, as well as the black color of the samples and the strong magnetic moment showed in contact with neodymium magnet, it can be stated that all the prepared MNPs contain magnetite as single crystalline phase.

#### 3.1.2. MNPs’ Thermal Behavior

Figure 4 exhibits the TG-DSC curves of both types of MNPs obtained by green synthesis from ethanolic extracts of *Camellia sinensis* L. (Figure 4A,B) and *Ocimum basilicum* L. (Figure 4C,D). 

All the samples present a similar thermal behavior. The endothermic effect recorded under 100 °C, accompanied with a mass loss on the TG curve might be attributed to water evaporation or decomposition of unstable bioactive compounds, which seems to be more obvious in the case of nanoparticles synthesized at 25 °C, according to the weight loss percentage. In the temperature range 200–400 °C, a strong exothermic effect was recorded in all the cases, accompanied by the highest mass loss. These large effects can be explained by the overlapping of the exothermic effect assigned to the oxidation of organic compounds located on the surface/pores of the MNPs. 

In the case of samples synthesized at 25 °C (Cs 25 and Ob 25), a small exothermic effect occurs at 517.1 °C (Cs 25) and at 512.3 °C (Ob 25), respectively, with a low mass loss of each—1.28% (Cs 25) and 1.13% (Ob 25). These effects could be related to the polymorph transition of maghemite to hematite as well as to the conversion of carbohydrates or/and amino acids/proteins into secondary metabolites. In the case of samples synthesized at 80 °C (Cs 80 and Ob 80), exothermic effects with low mass loss (under 1%) are also recorded, one at 599.7 °C (Cs 80) and the other at 609.1 °C. These effects are related to the carbon content oxidation originating from the organic biomolecules of the plant materials.

#### 3.1.3. MNPs’ Electron Microscopy Characterization

Figure 5 depicts the SEM and TEM images of the MNPs obtained by green synthesis from Cs (green tea) (A,B) and Ob (basil) (C,D) ethanolic extracts, at 25 °C and at 80 °C, as well as the chemical composition of both samples, revealed by EDX analysis.

Regarding the MNPs obtained from green tea extract, regardless the temperature at which the reaction took place, the morphology of the samples was almost the same. Additionally, a decrease in aggregation of MNPs and more singular, nearly spherical nanoparticles were observed when the reaction temperature was increased at 80 °C (Figure 5B). Moreover, compared to the MNPs obtained at 80 °C (Cs 80), the MNPs fabricated at 25 °C (Cs 25) appear with large amounts of bioactive compounds from green tea ethanolic extract. 

In terms of the elemental composition of MNPs determined by EDX, Fe and O appear as major components, alongside Cl, Si, and Ca in a neglected total weight percent (Figure 5A). These elements are present only when the reaction took place at 25 °C. Besides these elements, the EDX analysis also exhibits carbon. 

Regarding the MNPs obtained from basil extract, a difference can also be observed (Figure 5C,D). MNPs synthesized at 25 °C appear more agglomerated than those synthesized at 80 °C, but the spherical shape is conserved as well as the nanometric scale (under 10 nm). The EDX spectra revealed that when the synthesis occurs at 25 °C (Figure 5C), besides Fe, O, and C, also Si and Cl are present, in a total weight percentage higher than when the green tea ethanolic extract was used.

Based on the TEM and SEM images, the MNPs’ diameter (*D_EM_*) statistics were determined by using ImageJ software version number 1.53f [https://imagej.nih.gov/ij/, accessed on 30 October 2022]. The obtained *D_EM_* values are presented in Table 3. 

#### 3.1.4. MNPs’ Magnetic Measurements

The MNPs’ first magnetization and hysteresis curves are presented in Figure 6. The MNPs’ saturation magnetization (M_sat_), remanent magnetization (M_r_), coercive field (H_c_), and magnetic diameter (*D_m_*) are presented in Table 3.

At the surface of the MNPs exists a nonmagnetic layer so that the total physical diameter is *D_EM_*, which is higher than the MNPs’ magnetic diameter (*D_m_*). The thickness of the nonmagnetic layer can be quantified according to the equation below (Equation (3)) and the value is presented in Table 3:(3)$$\delta_{nm}=\frac{D_{EM}-D_{m}}{2}$$

The thickness of the nonmagnetic layer of Cs nanoparticles is about 70% smaller than that of the Ob nanoparticles, regardless of synthesis temperature. 

The saturation magnetization and the MNPs magnetic diameter are fairly well correlated. The saturation magnetization is higher in the samples prepared with green tea extract (Cs) and for 80 °C synthesis temperature. The samples synthesized at 25 °C show zero magnetic remanence.

### 3.2. Biological Assessment of MNPs Obtained via Green Synthesis (Cs 25, Cs 80 and Ob 25, Ob 80)

To provide a complex in vitro biological profile of the test samples (Cs 25, Cs 80 and Ob 25, Ob 80), multiple in vitro methods were performed in the current study by using one normal 3D human bronchial microtissue (EpiAirway^TM^, MatTek Corporation) and two different lung cancer cell lines (A549 and NCI-H460).

#### 3.2.1. Biosafety Profile Using EpiAirway^TM^ 3D In Vitro Microtissues

##### Acute Biosecurity Assessment through MTT Test at 24 h Post Exposure

The biosecurity level of the MNPs obtained by green synthesis (Cs 25, Cs 80 and Ob 25, Ob 80) was assessed on a human functional microtissue model containing bronchial cells (EpiAirway^TM^, MatTek Corporation), presenting good physiologic similarity with the human respiratory tract. The results obtained are presented in Figure 7.

Figure 7 presents the viability percentage of EpiAirway^TM^ 3D in vitro microtissues after treatment with the highest two test concentrations (300 µg/mL and 500 µg/mL) of MNPs obtained by green synthesis, starting from: (i) ethanolic extract of *Camellia sinensis* L., at 25 °C and 80 °C (Cs 25, Cs 80) and (ii) ethanolic extract of *Ocimum basilicum* L., at 25 °C and 80 °C (Ob 25 and Ob 80).

The MTT test revealed that after 24 h post treatment, none of the test samples induced an important viability impairment of the EpiAirway^TM^ 3D microtissues, the viability rates being above 80%.

##### Histological Aspects of 3D Respiratory Tissues Model

Figure 8 presents the morphological aspects of the sample-treated EpiAirway^TM^ 3D microtissues after exposure to the highest test concentration (500 µg/mL) of MNPs obtained by green synthesis starting from both ethanolic extract of *Camellia sinensis* L. and *Ocimum basilicum* L., at 25 °C and 80 °C (Cs 25, Cs 80—Figure 8A and Ob 25, Ob 80—Figure 8B).

For both samples, the Control consist in human-derived model with aspect of multilayered tracheal/bronchial like epithelial cells developed on a microporous membrane (HE, Ob. ×40) (Figure 8A,B Control). 

EpiAirway^TM^ 3D models treated with MNPs obtained from green tea at 25 °C showed a slightly denudated superficial layer; however, no denudation from the microporous membrane or loose epithelial junctions were observed (Figure 8A—Cs 25). Concerning the impact of sample Cs 80 obtained from green tea at 80 °C, surface epithelium has slight aspects of denudation from the microporous membrane (Figure 8A—Cs 80).

Figure 8B shows the biological effect of both samples Ob 25 and Ob 80 (obtained from basil ethanolic extract at 25 °C and 80 °C, via green synthesis), at a test concentration of 500 µg/mL on the multilayered respiratory tract epithelium. The sample Ob 25 induced a slight loose of the epithelial junctions of the multilayered respiratory tract epithelium (HE, Ob. ×40) (Figure 8B—Ob 25). However, the sample Ob 80 shows an area with no alteration of the main histological aspect (Figure 8B—Ob 80).

#### 3.2.2. Cell Viability Assessment on Tumorigenic A549 and NCI-H460 2D Cell Cultures

The Alamar blue test was performed to evaluate the cell viability percentage of the lung cancer cell lines (A549 and NCI-H460) after exposure to MNPs obtained by green synthesis (Cs 25, Cs 80 and Ob 25, Ob 80), for several intervals of time: 24, 48, 72, 96 h. The results obtained through the Alamar blue test are presented in Figure 9.

As presented in Figure 9, the cell viability rate of human lung cancer—A549 cells was not affected at 24 h post treatment with Cs 25 (concentrations 150–500 µg/mL)—the cells manifesting viabilities around 100%. After exposing the cells to Cs 80 for the same interval of time (24 h), cell viability percentages were decreasing in a concentration dependent-manner (150–500 µg/mL), the viable rate ranging between 100–80%, respectively. However, when the exposure time was increased to 48–96 h, the viable population of A549 cells decreased, with a significant drop being recorded between 24 h and 48 h post treatment, where the most important biological impact was presented by Cs 80 at concentration of 500 µg/mL, inducing a viability percentage of A549 cells between 71.5—70% (the viability decreasing in a time-dependent manner), while the Cs 25 induced a viability above 80%, under the same conditions (concentration of 500 µg/mL and 48–96 h exposure time). However, the effect induced by Ob 25 and Ob 80 on the A549 cell culture is more noxious, compared to the effect observed for Cs 25 and Cs 80, as the cell viability percentage was around 93–81% after exposure to Ob 25 for 24 h. Nevertheless, the same viability decrease impact was observed between 24 h and 48 h exposure time intervals, as the one presented above for Cs 25 and Cs 80. The viable A549 cell population elicited a higher decrement after exposure to Ob 25 and Ob 80 (from 48 h up to 96 h), compared to Cs 25 and Cs 80, the viability rate being around 67–63% for Ob 25 (500 µg/mL), while the viability percentage for Ob 80 (500 µg/mL) is ranging between 66–51%; the viability percentage is decreasing depending on the incubation time (from 48 h to 96 h).

The viability percentage manifested by the large human lung carcinoma—NCI-H460 cells after treatment with Cs 25 and Cs 80 at three different concentrations (150, 300, 500 µg/mL) for four intervals of time, from 24 h up to 96 h. Still, the viability rate of the cells is only slightly affected by the exposure to these samples. Two of the most important decreases in cell viability rate were observed after the cells were treated with Cs 25 (concentration of 500 µg/mL) for 72 h and 96 h, thus obtaining a viability of 86.32% and 80.07%, respectively. However, the rest of the samples induced cell viabilities around 90%. 

The biological effect induced by Ob 25 and Ob 80 on large human lung carcinoma—NCI-H460 cells shows a more pronounced viability decrement compared to the effect induced by Cs 25 and Cs 80 on NCI-H460 cells; in this case, Ob 25 (500 µg/mL) induced viability rates of 86.77%, 72.10%, and 71.98% after 48 h, 72 h and 96 h, respectively; while Ob 80 induced viability percentages of 83.88%, 70.14%, and 66.33% under the same conditions (concentration of 500 µg/mL and time interval between 48–96 h). Nevertheless, it can be easily observed that the most important drop of the cell viability was recorded between 48 h and 72 h, a different pattern compared to the one observed for Cs 25, Cs 80 and Ob 25, Ob 80 samples, where the most significant drop was recorded between 24 h and 48 h. These aspects indicate that human lung carcinoma—A549 cells are more sensible compared to NCI-H460 cells when exposed to MNPs obtained by green synthesis (Cs 25, Cs 80 and Ob 25, Ob 80). 

#### 3.2.3. Cytotoxic Potential on A549 and NCI-H460 2D Cell Cultures

To better understand the biological impact of test samples, the LDH release method was employed after 96 h post treatment. The time interval of 96 h was chosen based on the results obtained via Almar blue test, where the most reduced cell viability rates were observed.

The results showed that the most cytotoxic effect was manifested by Cs 25 and Cs 80 on human lung carcinoma—A549 cells when applied at concentrations of 300 µg/mL and 500 µg/mL (Figure 10A). In this case, the A549 cells releasing extracellularly the highest amount of LDH—a percentage of 21.29% and 23.06% was recorded after the cells were treated with Cs 25 (at concentrations of 300 µg/mL and 500 µg/mL), while cytotoxic rates of 24.08% and 25.2% were obtained after treating the A549 cells with Cs 80 at the same concentrations (300 µg/mL and 500 µg/mL). Ob 25 and Ob 80 induced lower cytotoxic rates on A549 cells (below 18.5%).

## 4. Discussion

Advances in the synthesis and engineering of MNPs offer more specific alternatives for diagnosis and treatment of various cancer types [19,35]. The synthesis method plays an essential role in preparation of MNPs with tailored features, suitable for biomedical applications. Magnetic iron oxide nanoparticles can be synthesized by physical (i.e., laser ablation; spray pyrolysis; gas-phase deposition) [36,37,38,39], chemical (i.e., coprecipitation; thermal decomposition; sonochemical and microemulsion; microwave-assisted; hydrothermal and solvothermal; sol-gel; electrochemical; combustion) [40,41,42,43,44,45,46,47,48,49,50], and biological methods [51,52,53,54,55,56,57,58]. The physicochemical methods mentioned above present different disadvantages for the MNPs fabrication. For instance, the physical methods are expensive, complicated and require subsequent chemical reactions to obtain solid NPs followed by a heat treatment [48,59]. Concerning the chemical methods, even if coprecipitation is by far the simplest and used method, the broad size distribution, particles’ size and shape are still the main issues hard to control [60,61]. The thermal decomposition of organometallic precursors method needs long reaction time; also, it is a complicated method which involves the use of toxic solvents and surfactants with potential hazards (environmental toxicity, cytotoxicity and carcinogenity) which are not compatible with biomedical applications [48,59,62,63]. Regarding the hydrothermal/solvothermal method, the drawbacks consist in high cost of the equipment, the synthesis process is impossible to monitor due to the need of high pressure and the reaction time could takes several days [48,60]. The disadvantages presented by the sonochemical method are as follows: the bad control of particles shape and energy inefficiency [64,65]. The microemulsion method requires large amounts of solvent and multiple washing stages to remove the excess of surfactant [48,60,66]. As regards the electrochemical method, this requires the use of specialized equipment and even days until the MNPs’ formation [67,68]. 

Thus, developing a clean, simple, single step, cost-effective, eco-friendly, biological compatible, and benign green method to synthesize MNPs represents a major important topic [69]. The green method, which in recent years represents an attractive alternative to conventional methods for synthesizing MNPs, implies the use of unicellular and multicellular biological entities, such as plants, viruses, fungus, bacteria, enzymes, and biomolecules, yeasts and actinomycetes [70,71,72,73]. From all biological entities, plants are the most used in green synthesis due to the easy process of obtaining, reduced costs, large-scale production, and environmental benign [74]. The whole plant or different parts of plants (leaves, fruits, fruit peels, stems, roots, seeds, etc.) can participate in the synthesis of MNPs, due to the notable amounts of phytochemicals which can act as reducing and stabilizing agents in the synthesis process. Moreover, in the reaction medium, the phytochemicals may reduce metal cations in a single step, from their mono/divalent oxidation states to zero valent state, i.e., to metal nanoparticles. The biochemical reaction is easy to conduct at room temperature or at higher temperature and it is completed within a few minutes. 

In accordance with the above, the present study aimed to obtain and characterize MNPs by green synthesis method, starting from two ethanolic extracts, based on leaves of *Camellia sinensis* L. and *Ocimum basilicum* L. The novelty of the work consists in the fact that no study has previously reported the potential application of such biosynthesized MNPs’, from green tea and basil, in lung cancer. The present work being the first research on this topic—synthesis of MNPs by using green tea and basil ethanolic extracts as reducing agents, followed by a complex in vitro screening. 

The outcomes showed that, in terms of phase composition, there are no significant differences among the synthetized MNPs, and the results are in good agreement with the scientific literature [75,76,77,78,79].

The crystallite size of the obtained MNPs, seems to be influenced by the extract type (basil—Ob or green tea—Cs) as well as the reaction synthesis temperature (25 °C and 80 °C). For instance, in the case of MNPs obtained using basil extract (Ob), the crystallites’ size decreases from 8 nm (Ob 80) to 6 nm (Ob 25) as the reaction synthesis temperature decreases from 80 °C to 25 °C. As concerns the MNPs prepared using green tea (Cs), the crystallites’ size is the same—7 nm, regardless the reaction synthesis temperature (25 °C or 80 °C). One of the parameters which could influence this aspect might be the type and/or the plant extract concentration, as it was reported that the use of various plant extracts with different types of phytocompounds affects the size and shape of the MNPs [80]. Moreover, the volume ratio of the extracting solvent to plant material as well as the extraction temperature are two essential parameters during the MNPs’ synthesis with tailorable morphological features. The extraction temperature in the case of green tea (70 °C) as well as the extraction time (1 h/cycle), may have been too high, and certain phytocompounds may have been degraded [28]. This aspect could have had an influence on the MNPs’ growth when the synthesis was carried out at 80 °C. 

Regarding the MNPs’ thermal behavior, at 25 °C, a small exothermic effect was recorded, in both cases (at 517.1 °C for Cs 25 and at 512.3 °C for Ob 25, respectively), both with a low mass loss (Figure 4A,C). These effects could be related to the polymorph transition of maghemite to hematite as well as to the conversion of carbohydrates or/and amino acids/proteins into secondary metabolites. Moreover, due to their additional water molecules, the oxidation process can occur. This could be the explanation why at 25 °C, on DSC curves, the exothermic processes were recorded after 500 °C; most likely, these two samples contain also γ-Fe_2_O_3_ beside Fe_3_O_4_ and, after 500 °C, the polymorph transition of maghemite to hematite took place. As concerns the samples synthesized at 80 °C, exothermic effects with low mass loss are also recorded, but at higher temperatures (599.7 °C in the case of Cs 80 NPs and 609.1 °C in the case of Ob 80). These effects are related to the carbon content oxidation originated from the organic biomolecules of plant materials. Thus, one can affirm that the incorporation of bioactive compounds of the plant extracts on the surface or in the pores of MNPs occurred [81]. Therefore, considering the results obtained, and corroborating with those from the XRD analysis, it can be stated that the synthesis of MNPs at 25 °C, starting from the ethanolic extracts of green tea and basil, leads to the formation of a mixture of magnetite (as the predominant phase) and maghemite (probably in traces). In the case of the synthesis at 80 °C, it may be assumed that only magnetite was obtained as a single main phase, the present results being in good agreement with scientific literature [79,82,83,84]. 

The MNPs’ electron microscopy characterization revealed that at 25 °C (Cs 25), the MNPs possess larger amounts of bioactive compounds from green tea ethanolic extract. This outcome can be explained by the fact that the polyphenols and/or caffeine concentrations from green tea extract play a significant role in the formation of MNPs’ final structure, since they are acting as reducing and capping agents. In addition, it was stated that the secondary metabolites from plant extracts with functional groups such as C=C; O-H; N-H; C-N; C-H and COO-, which might be micro- and macro-biomolecules, are the main factors for the MNPs’ biosynthesis [85]. It is quite possible that, at 25 °C, these micro- and macro-biomolecules did not degrade and remained in larger quantities on the surface or in the pores of the preformed MNPs. The EDX analysis revealed that Fe and O are the major elements of the MNPs, however, traces of Cl, Si, Ca were also recorded (Figure 5A). Since it was stated that some elements were found in traces in chemical composition of green tea [86], this aspect may be the reason why they were detected; but, to the same extent, these elements can come from the growing soil of the plant or an unreacted metal precursor (FeCl_3_). The presence of carbon on EDX spectra highlights the fact that this element may originate from the tape used as grid support for MNPs’ immobilization, as well as to the residual organic biomolecules of plants’ ethanolic extract, which occurs in a high amount when the reactions take place at 80 °C (Figure 5B,D)—in agreement with the TG-DSC curves (Figure 4B,D). Nevertheless, the MNPs’ fabrication starting from green tea ethanolic extract, lead to nearly similar spherical shape and particles size under 10 nm, regardless the temperature of the reaction medium. Our results are in accordance with literature data [87,88,89,90]. Comparing both types of MNPs, those obtained from green tea and basil extracts, one can notice that the total weight percentage of Fe decreases with a temperature increase. Most probably, at 80 °C, the formation of MNPs occurs fast, therefore, reduction of the metallic precursor occurs faster. While at 25 °C, the total formation of MNPs takes place over time and, perhaps, 2 h of reaction is insufficient. This affirmation is in agreement with the statement reported by Cruz and co-workers [91], which affirmed that with the increase of reaction temperature, the efficiency of the metal ion reduction increases. According to literature data, a higher amount of phytocompounds present in the reaction medium lead to the obtaining of small nanoparticles [92], but, at the same time, the faster the MNPs’ formation is, the smaller MNPs are obtained [57]. In our case, the MNPs’ formation was faster, but at higher temperature (80 °C), while synthesis at 25 °C seems to had influence on the nuclei’s growth. It is also known that a high concentration of polyphenols in the reaction medium allows to obtain nanoparticles with narrow size [93]. 

Regarding the magnetic features of the MNP samples, as presented in Table 3, the MNPs’ magnetic diameter (*D_m_*) is smaller than the physical diameter (*D_EM_*), the difference can be explained by the presence of a nonmagnetic layer at the surface of the MNPs which is caused by the loss of spin correlations at the surface [94], oxidation, and chemical interactions with surface-adsorbed molecules [95]. Since the thickness of the nonmagnetic layer of Cs nanoparticles is 70% smaller than the one with the Ob nanoparticles, this aspect may indicate that Cs is a more effective antioxidant than Ob.

However, since the newly synthesized green MNPs have been developed for biological applications, a complex in vitro screening by using different in vitro techniques was mandatory to establish their biological profile. In this regard, one normal 3D human airway epithelium (EpiAirway^TM^, MatTek Corporation) and two morphologically different lung cancer 2D monolayers (A549 and NCI-H460) were employed. The 3D bronchial microtissues were selected in the current study as they present a mucociliary epithelium and an active mucin secretion, thus reproducing under in vitro conditions the normal physiological parameters of the human airway epithelia [25,26,27,96]. 

On the other hand, since the potential toxicity of iron oxide NPs represents a debatable topic, the scientific literature reporting studies that sustain the benign feature of these MNPs, whereas others sustain their toxicological potential, from inflammatory reactions to genotoxic effects [97] or even homeostasis impairment when high concentrations of free iron ions are accumulated within a specific organ [98]. To provide reliable data of the biosafety level of the newly synthesized MNPs represents one of the main goals of the current study; thereby, the EpiAirway^TM^ models were mandatory to establish the biosecurity level of the samples by employing in our screening microtissues that could reveal data as close as possible to those that may occur in vivo (under normal physiological conditions).

The results (Figure 7) showed that the viability of the 3D microtissues was above 80% after exposure to the highest test concentration (500 µg/mL) of all green MNPs (Cs 25, Cs 80 and Ob 25, Ob 80). Thus, according to ISO Standard 10993-5:2009—the standard for the biological assessment of medical devices [99], a test sample is considered cytotoxic if inducing a viability rate under 70%; therefore, MNPs obtained in the present study (Cs 25, Cs 80 and Ob 25, Ob 80) are not considered toxic for bronchial epithelia when applied at concentration of 500 µg/mL. In addition, the viability data are sustained by the histological analysis of the microtissues (Figure 8A,B), where no important histological changes were recorded.

To obtain an overview picture of the anti-tumorigenic activity of MNPs obtained by green synthesis (Cs 25, Cs 80 and Ob 25, Ob 80), the Alamar blue test was performed on two different human lung carcinoma—A549 and NCI-H460 cells, by employing 4 intervals of time: 24, 48, 72, 96 h. 

As presented in Figure 9, a difference of approximately 10% of the cell viability rates were recorded between CS 25 and Cs 80 (when applied on A549 cells), a fact that might be related to the magnetic remanence features manifested by Cs 80 versus Cs 25, which did not elicit remanent magnetic forces, as presented in Table 3. In addition, a higher cell viability decrease was induced by Ob 25 and Ob 80, versus Cs 25 and Cs 80, which might be related to the higher nonmagnetic layer observed on the surface of Ob samples, compared to Cs-derived MNPs, as presented in Table 3. This aspect may be correlated to a higher deposit of phytocompounds from the Ob ethanolic extract on MNPs surface. In addition, the anti-tumorigenic activity of Ob on lung cancer cells is well known and is attributed especially to the ursolic acid content, by inducing mitochondrial-related apoptosis as described by Aminian et al. [100], whereas, according to Dwivedi et al. [101], the cytotoxic effect induced by iron oxide NPs on A549 is closely related to the interference of the functional groups of iron oxide NPs with the intracellular proteins of A549 cells. In addition, the same study [101] sustained that high concentrations of iron NPs induce cellular stress which is responsible for morphological alterations of A549 cells. This explanation is also sustained by our results (Appendix A), where concentration above 300 µg/mL of Cs 25 induced morphological changes of A549 monolayer, followed by cellular detachment.

Nevertheless, several other studies [22,102,103] reported good anticancer activity of iron nanoparticles on human lung cancer A549 cell line, results that corroborate with the ones obtained in the present study.

Regarding the antitumoral activity manifested by MNPs obtained by green synthesis (Cs 25, Cs 80 and Ob 25, Ob 80) on the two human lung carcinoma cell lines: A549 and NCI-H460, the Alamar blue test indicated that the viability percentage of A549 cell population was more affected compared to the viability of NCI-H460 cells (Figure 9). In other words, the A549 cell culture is more sensible to test sample treatment than NCI-H460 cells. This particular phenomenon may be explained by the high difference of the proliferation rate between the two cell lines: the A549 monolayer manifesting a tumor proliferation rate at half compared with NCI-H460 cells [104].

Although the Alamar blue test showed that Ob 25 and Ob 80 induced the most reduced cell viability percentages, the cytotoxic percentages via LDH release test did not endorse the results (Figure 10). This aspect may be related to the fact that the samples (Ob 25 and Ob 80) interfered with cell proliferation (as presented in the cell morphology assessment section—Appendix A); thus, the cells cannot release a higher amount of LHD since they are in a cell cycle arrest phase, therefore, the amount of LDH released is lower. This phenomenon was also experimented by Ghițu et al. [105] and was correlated with the cell cycle arrest effect of the samples.

Nevertheless, a large proportion of the cytotoxic events induced by iron oxide MNPs are closely related to the physicochemical features of the particles which further influence the entry route of the NPs into the cells, mainly through micro-pinocytosis and endocytosis processes and leads to the intracellular accumulation [106]; an important cascade of factors which are responsible for the overall toxicity of the MNPs and which should be further investigated in order to obtain an insight into the mechanism of action of the MNPs. 

## 5. Conclusions

In the present study, a green one-step method was employed for the synthesis of magnetic iron oxide nanoparticles, using plant materials (green tea and basil) as reducing and capping agents. The physicochemical analysis showed that the samples prepared at 25 °C contain a mixture of Fe_3_O_4_ and γ-Fe_2_O_3_ nanoparticles, while the synthesis at 80 °C led to the formation of Fe_3_O_4_ nanoparticles as a unique phase. The electron microscopy exhibited that the formed MNPs have nearly spherical shape with narrow size under 8 nm. Moreover, the MNPs’ diameter statistics, determined from TEM and SEM images, is in accordance with their crystallite sizes, determined by XRD analysis and with their magnetic diameters. The magnetic measurements demonstrated the strong saturation magnetization, around 60 emu/g in the case of the samples prepared from green tea ethanolic extract, which makes these MNPs being suitable also for hyperthermia applications as well as for drug delivery. 

The acute biosecurity screening of MNPs showed good results, the EpiAirway^TM^ microtissues manifesting viabilities above 80% with no significant histopathological changes. In addition, the anticancer potential of the samples on lung cancer cells was evaluated, the data revealing that the MNPs are more active on A549 cells compared to NCI-H460 cells. 

Taking into account all the promising features presented above, the newly synthesized MNPs can be considered suitable candidates for development of nanotechnology-enabled formulations for lung cancer therapy.

## Figures and Tables

**Figure 1 pharmaceutics-15-00002-f001:**
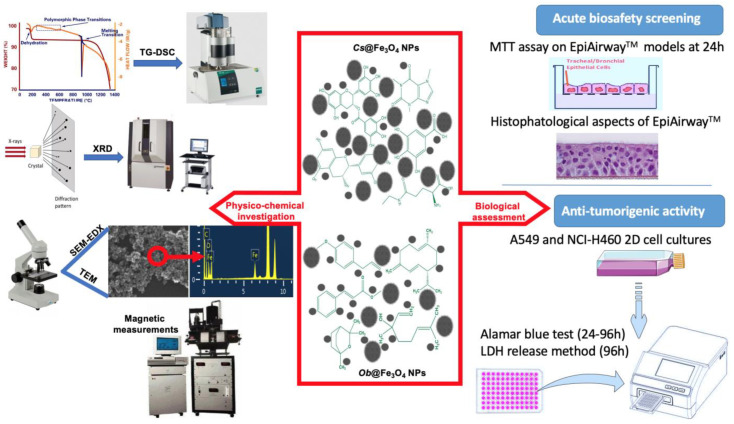
Schematic representation of the methodology section.

**Figure 2 pharmaceutics-15-00002-f002:**
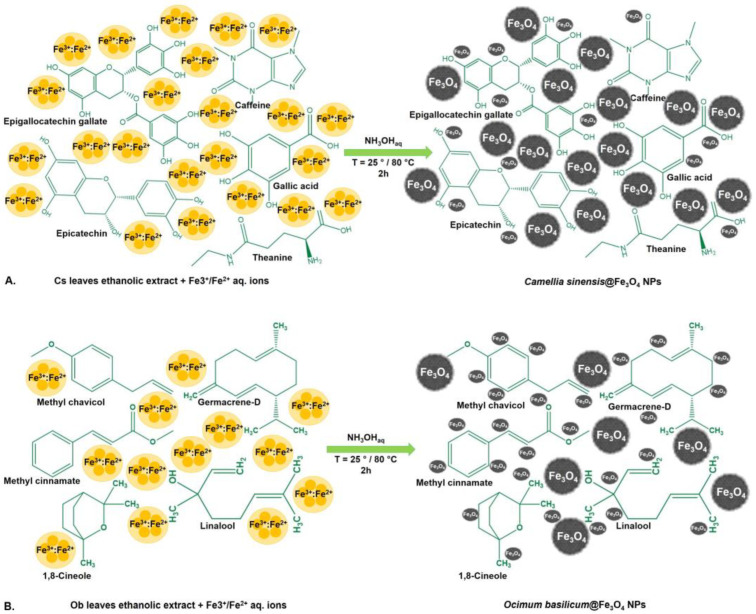
Schematic representation of synthesized MNPs interactions with activated functional groups of *Camellia sinensis* (**A**) and *Ocimum basilicum* (**B**) leaves ethanolic extracts.

**Figure 3 pharmaceutics-15-00002-f003:**
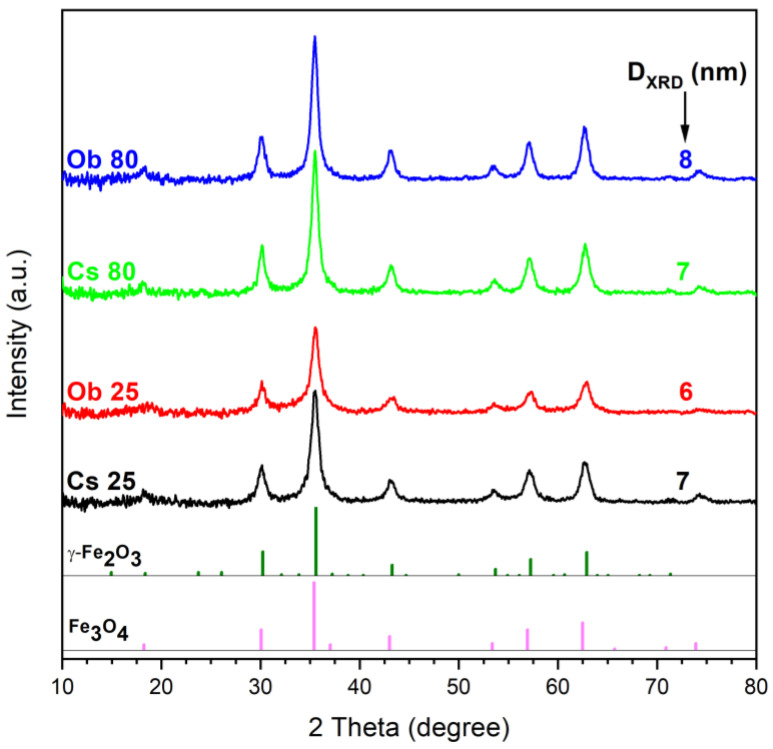
XRD patterns of the MNPs obtained by green synthesis from Cs (green tea) and Ob (basil) ethanolic extracts, at 25 °C and 80 °C.

**Figure 4 pharmaceutics-15-00002-f004:**
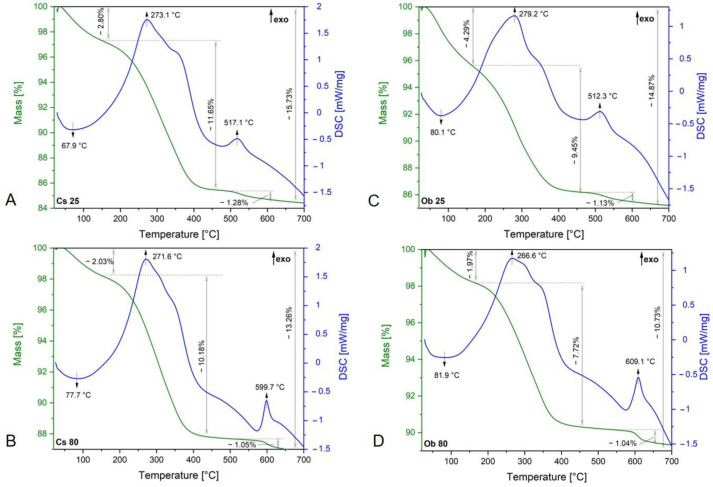
TG−DSC curves of MNPs obtained by green synthesis from Cs (green tea) and Ob (basil) ethanolic extracts, at 25 °C (**A**,**C**) and at 80 °C (**B**,**D**).

**Figure 5 pharmaceutics-15-00002-f005:**
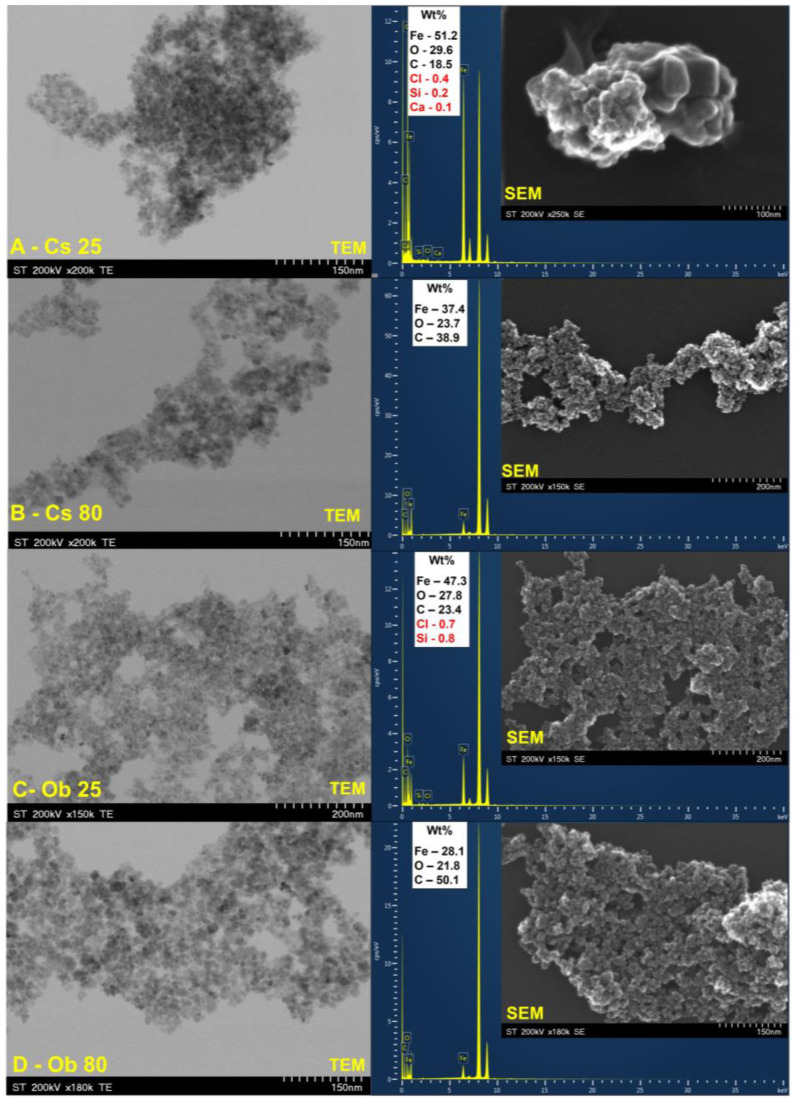
TEM and SEM images alongside EDX spectrum of MNPs obtained by green synthesis from Cs ethanolic extract at 25 °C (**A**) and at 80 °C (**B**), as well as from Ob ethanolic extract at 25 °C (**C**) and at 80 °C (**D**).

**Figure 6 pharmaceutics-15-00002-f006:**
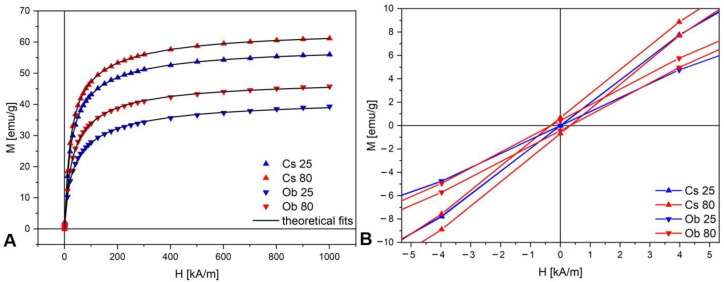
(**A**) First magnetization curves of the samples, and (**B**) hysteresis curves of the samples (zoom).

**Figure 7 pharmaceutics-15-00002-f007:**
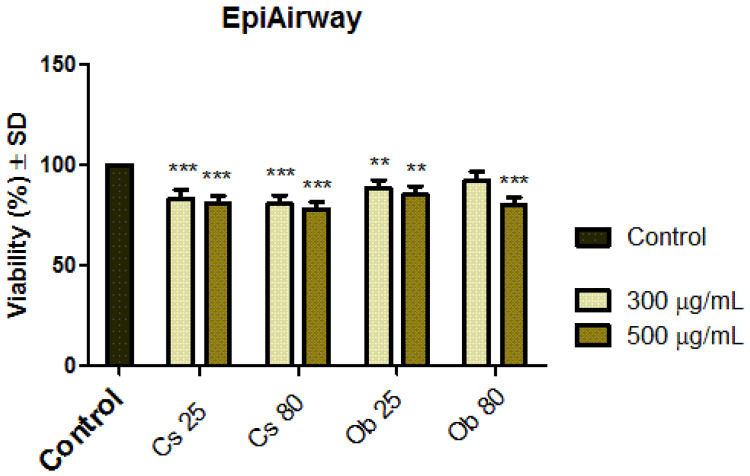
Viability rate of EpiAirway^TM^ 3D in vitro microtissues after treatment with Cs 25, Cs 80, Ob 25 and Ob 80 at the highest two test concentrations of 300 µg/mL and 500 µg/mL for intervals of 24 h. The results represent the mean values of three independent experiments ± standard deviation (SD). One-way ANOVA analysis was applied to determine the statistical differences followed by Dunnett’s comparisons test (** *p* < 0.01; *** *p* < 0.001).

**Figure 8 pharmaceutics-15-00002-f008:**
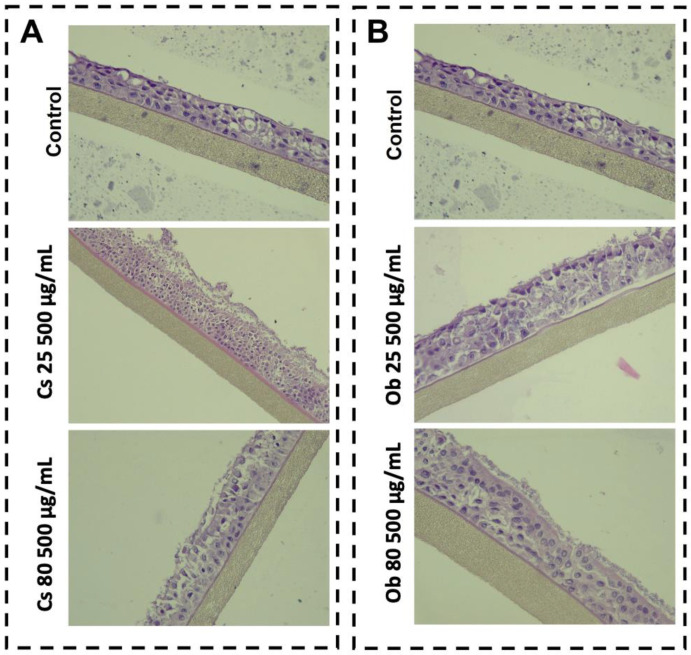
Morphological aspects induced by MNPs based on green tea (**A**) and basil (**B**), on bronchial respiratory tissues.

**Figure 9 pharmaceutics-15-00002-f009:**
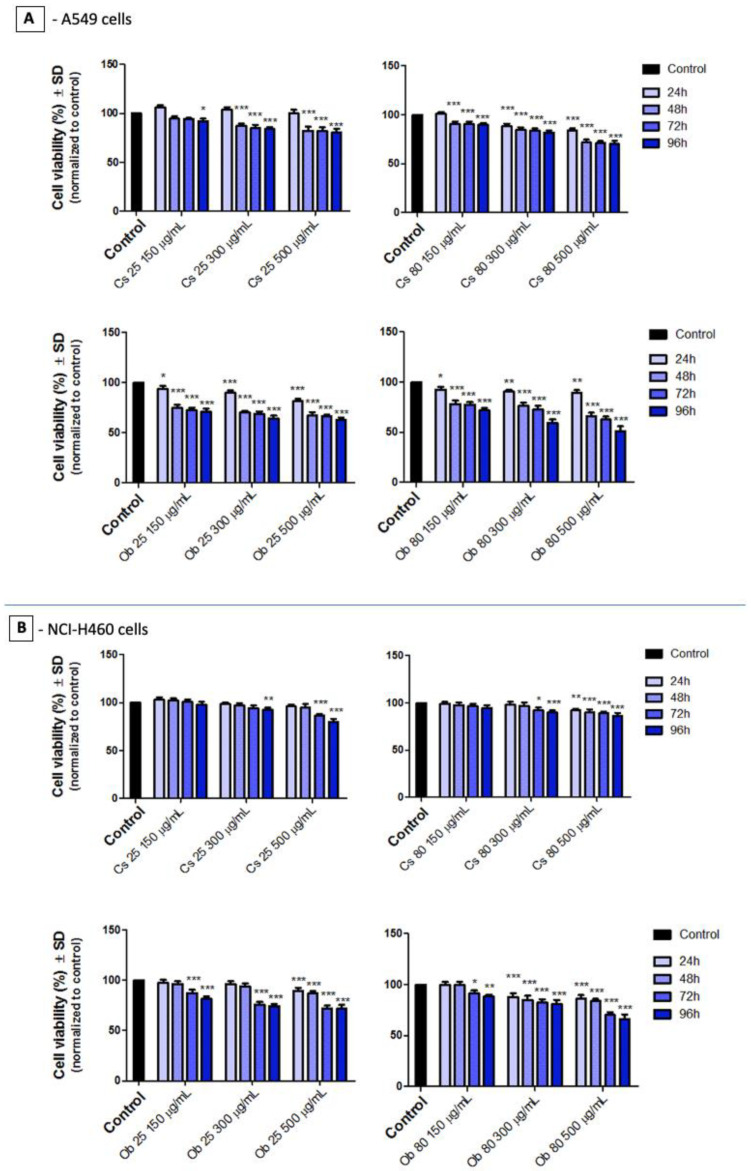
Cell viability percentage of: (**A**) human lung carcinoma—A549 cells and (**B**) human lung carcinoma—NCI-H460 cells after exposure to test samples (Cs 25, Cs 80 and Ob 25, Ob 80) at three different concentrations (150, 300, 500 µg/mL) for intervals of 24, 48, 72, and 96 h. The results represent the mean values of three independent experiments ± standard deviation (SD). One-way ANOVA analysis was applied to determine the statistical differences followed by Dunnett’s comparisons test (* *p* < 0.1; ** *p* < 0.01; *** *p* < 0.001).

**Figure 10 pharmaceutics-15-00002-f010:**
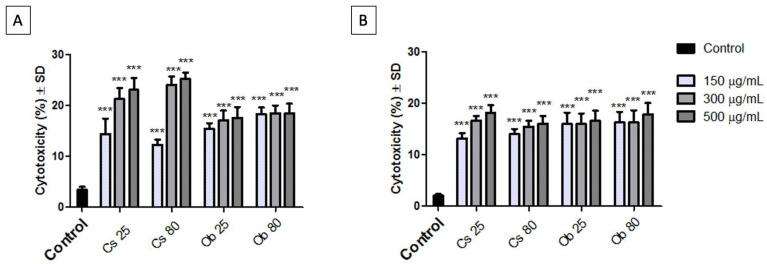
Cytotoxicity rate of (**A**) human lung carcinoma—A549 cells and (**B**) human lung carcinoma—NCI-H460 cells post exposure to Cs 25, Cs 80, Ob 25 and Ob 80 at three different concentrations (150, 300, 500 µg/mL) for an interval of 96 h. The results represent the mean values of three independent experiments ± standard deviation (SD). One-way ANOVA analysis was applied to determine the statistical differences followed by Dunnett’s comparisons test (*** *p* < 0.001).

**Table 1 pharmaceutics-15-00002-t001:** The synthesis conditions used for MNPs’ preparation.

No.	MNPs Annotation	Raw Materials	Working Temperature (°C)	Precipitation Agent	Observations
1.	Cs 25	Fe^3+^:Fe^2+^ = 2:1 6 mg/mL of plant extract metal precursor to plant extract volume ratio = 1:1	25	NH_4_OH 25%	*Camellia sinensis* leaves ethanolic extract
2.	Cs 80		80		
3.	Ob 25		25		*Ocimum basilicum* leaves ethanolic extract
4.	Ob 80		80		
Total quantity of the MNPs obtained by green synthesis (g)					
Cs 25		1.36231	Ob 25		1.36877
Cs 80		1.34348	Ob 80		1.29215

**Table 2 pharmaceutics-15-00002-t002:** The 2*θ* values of the obtained MNPs as compared to 2*θ* values of Fe_3_O_4_ and γ-Fe_2_O_3_ from the International Centre for Diffraction Data Powder Diffraction File.

2*θ* Values		
Magnetic Nanoparticles	Magnetite—Fe_3_O_4_ (PDF: 190629)	Maghemite—γ-Fe_2_O_3_ (PDF: 391346)
18.18°	18.269	18.384
30.10°	30.095	30.241
35.49°	35.422	35.63
43.16°	43.052	43.284
53.47°	53.391	53.733
57.12°	56.942	57.271
62.72°	62.515	62.925
71.07°	70.924	71.376
74.22	73.948	-

**Table 3 pharmaceutics-15-00002-t003:** Magnetic characteristics of the samples.

No.	Sample Denomination	M_sat_@1MA/m (emu/g)	M_r_/M_sat_ (%)	H_c_ (kA/m)	*D_m_* (nm)	*D_EM_* (nm)	δ_nm_ (nm)	D_XRD_ (nm)
1	Cs 25	56.0	0	0	6.3 ± 2.3	7.2 ± 1.9	0.45	7
2	Cs 80	61.2	1.1	0.45	6.5 ± 2.2	7.5 ± 1.9	0.50	7
3	Ob 25	38.9	0	0	4.7 ± 2.3	6.3 ± 2.1	0.75	6
4	Ob 80	45.8	0.9	0.45	5.9 ± 2.2	7.3 ± 1.8	0.70	8

## Data Availability

Authors can provide raw data upon request.

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
