# Peer review of "Biologic Impact of Green Synthetized Magnetic Iron Oxide Nanoparticles on Two Different Lung Tumorigenic Monolayers and a 3D Normal Bronchial Model—EpiAirwayTM Microtissue"

_pharmaceutics, 2022, doi:10.3390/pharmaceutics15010002_

Round 1

Reviewer 1 Report

The authors present a research study regarding the development and evaluation of magnetic iron oxide nanoparticles with in vitro testing against lung tumorigenic monolayers and a 3D normal bronchial model. The MS is of two parts. One describes the synthesis and characterization of the MPNs and the second their in vitro impact as potential therapeutic vehicles for 2 lung cancer cell lines and a 3D tissue model of the human airway. This creates a problem for the MS that it has many things to show and discuss. The authors tried to organize their work and present as much as possible results they can but this makes the MS difficult to read sometimes. The manuscript needs major edits to provide key-points and leave the rest to supplementary or in the appendix. What was the aim of the study and how the issue is approached? What is new? How the Fe-MPN work in the cells? What are their potential toxicities? What are the desired properties for them?

 I suggest that the MS to be heavily revised leaving out unnecessary parts and focus on the main aspects of this study. 

Title: 

The term green is kind of strange maybe plant oriented, or green synthetized could be better.

Introduction: 

The introduction section must be heavily edited with a focus on the subject of this study. For example the authors spent a big section to describe epidemiological data of lung cancer (without seperating subtypes etc.) which are very general and well known. This section could be a lot shorter.  

For exaple: Please edit lines 40-55 to one paragraph and provide relative references for Romania

Also lines 40-63 can be further edited to be shorter. They are not so relative with the research but introductory and very general for lung cancer. 

Line 95. desired maybe not a proper word

Line 96. What NSCLC stands for? I guess non-small cell lung cancer. Also MNPs developed against squamous cell carcinoma, large cell carcinoma, or adenocarcinoma? 

Line 76. All previous facts could be ommitted and the introduction started from here. 

Lines 103-134 could move to the discussion section. 

Lines 135-144. What is missing regarding MPNs that this study will try to accomplish? 

Methods:

A figure of the overall methodology would be helpful 

2.5.1-2.5.4 for each section the contained text could be organized as one paragraph. 

Why for the cytotoxicity assays 3 time points were used whereas for EpiAirway only 24h? 

Results

Same fore 3.1 as above. The paragraphs are not so good organized withi the MS. 

Results are presened with commenting so it is like combining results and discussion together. Usually results are simple presentation of facts. Phrases like "At first glance it can be seen that there are no significant differences between all four types of magnetic nanoparticles. One can observe in all four cases ..." can seem like discussion so relative references are needed. Moreover the use of phrases like "at first glance" "one can observe" lack scientific formality. Please revise the text accordingly. It goes like that up to line 500. 

Figure 7-8. I think could be one figure. The same goes with 9-10. Also too many graphs maybe the authors use some representative (or best results) and put the rest in supplementary. There is also a comment that needs to be addressed if 3 samples are enough for performing statistical comparison such as with Dunnet's test especially if all values are normalized against the control. 

Since the highest cytoxicity is observed in 96h, why the authors did not present results for 96h in EpicAirway? It could point out the long-term low toxicity of their MPNs

Figures 13-14 could be one figure and text be edited. 

Discussion. 

Since they are presenting a method for developing MPNs it woudl be proper to further elaborate on their results regarding previously published studies (line s690-691). For example, up to line 724 we do not see many literature comparisons. 

Discussion could be further edited and improved regarding the presentation of key points regarding this work.

Final remark. There are some works that could be mentioned in the introduction section or during discussion regarding the orientation of nanomedicine technologies in cancer research and therapeutics. For example: 

1) Hanoglu SB, Man E, Harmanci D, et al. Magnetic Nanoparticle-Based Electrochemical Sensing Platform Using Ferrocene-Labelled Peptide Nucleic Acid for the Early Diagnosis of Colorectal Cancer. Biosensors (Basel). 2022;12(9):736. Published 2022 Sep 7. doi:10.3390/bios12090736

2) Abu-Serie MM, Abdelfattah EZA. Anti-metastatic breast cancer potential of novel nanocomplexes of diethyldithiocarbamate and green chemically synthesized iron oxide nanoparticles. Int J Pharm. 2022;627:122208. doi:10.1016/j.ijpharm.2022.122208

3) Curtis LT, Frieboes HB. The Tumor Microenvironment as a Barrier to Cancer Nanotherapy. Adv Exp Med Biol. 2016;936:165-190. doi:10.1007/978-3-319-42023-3_9

4) Vizirianakis IS, Mystridis GA, Avgoustakis K, Fatouros DG, Spanakis M. Enabling personalized cancer medicine decisions: The challenging pharmacological approach of PBPK models for nanomedicine and pharmacogenomics (Review). Oncol Rep. 2016;35(4):1891-1904. doi:10.3892/or.2016.4575

5) Tarawneh SFA, Dahmash EZ, Alyami H, et al. Mechanistic Modelling of Targeted Pulmonary Delivery of Dactinomycin Iron Oxide Loaded Nanoparticles for Lung Cancer Therapy [published online ahead of print, 2022 Nov 23]. Pharm Dev Technol. 2022;1-35. doi:10.1080/10837450.2022.2152047

Reviewer 2 Report

This manuscript reports the green synthesis of magnetic iron oxide nanoparticles and its characterization in vitro.  Generally, the results suggested that the nanoparticles were successfully synthesized, and their biosafety profiles were good.  Specific comments:

1.      The manuscript is relatively lengthy, and some sections can be significantly condensed.  For example, introduction can be shortened.

2.      Figure 4A, 4B, 5C, and 5D can be combined as Figure 4 that includes subfigures 4A, 4B, 4C, and 4D.

3.      Figure 7 to Figure 10 can be combined as one figure (Figure 6) that includes subfigures. 

4.      Figure 11 and Figure 12 can be combined as one figure that includes subfigures.

5.      Figure 13 and Figure 14 can be combined as one figure that includes subfigures.

6.      There are many grammatical errors.  Some examples are: ‘it was used ethylic alcohol’ (line 160), ‘handle grounded’ (line 174), ‘is closed to the one’ (line 300), ‘In Figure 2 are depicted’ (line 379).

7.      Total 114 references are cited.  The number of references can be reduced.

Round 2

Reviewer 1 Report

The authors presented an updated version of their work and addressed all the points made from the initial review evaluation. The manuscript can be further processed.